# Deep Learning-Based Estimation of Crop Biophysical Parameters Using Multi-Source and Multi-Temporal Remote Sensing Observations

**Hazhir Bahrami [1,*], Saeid Homayouni [2], Abdolreza Safari [1], Sayeh Mirzaei [3], Masoud Mahdianpari [4,5] and Omid Reisi-Gahrouei [6]**

1   School of Surveying and Geospatial Engineering, University of Tehran, Tehran 6791986363, Iran; asafari@ut.ac.ir
2   Centre Eau Terre Environnement, Institut National de la Recherche Scientifique, Québec, QC G1K 9A9, Canada; saeid.homayouni@ete.inrs.ca
3   School of Engineering Science, College of Engineering, University of Tehran, Tehran 6215845762, Iran; s.mirzaei@ut.ac.ir
4   C-CORE, St. John's, NL A1B 3X5, Canada; masoud.mahdianpari@c-core.ca
5   Department of Electrical and Computer Engineering, Memorial University of Newfoundland, St. John's, NL A1C 3S7, Canada
6   Department of Geomatics Science, Laval University, Québec, QC G1V 0A6, Canada; Omid.reisi-gahrouei.1@ulaval.ca
*   Correspondence: bahramihazhir@ut.ac.ir

**Abstract:** Remote sensing data are considered as one of the primary data sources for precise agriculture. Several studies have demonstrated the excellent capability of radar and optical imagery for crop mapping and biophysical parameter estimation. This paper aims at modeling the crop biophysical parameters, e.g., Leaf Area Index (LAI) and biomass, using a combination of radar and optical Earth observations. We extracted several radar features from polarimetric Synthetic Aperture Radar (SAR) data and Vegetation Indices (VIs) from optical images to model crops' LAI and dry biomass. Then, the mutual correlations between these features and Random Forest feature importance were calculated. We considered two scenarios to estimate crop parameters. First, Machine Learning (ML) algorithms, e.g., Support Vector Regression (SVR), Random Forest (RF), Gradient Boosting (GB), and Extreme Gradient Boosting (XGB), were utilized to estimate two crop biophysical parameters. To this end, crops' dry biomass and LAI were estimated using three input data; (1) SAR polarimetric features; (2) spectral VIs; (3) integrating both SAR and optical features. Second, a deep artificial neural network was created. These input data were fed to the mentioned algorithms and evaluated using the in-situ measurements. These observations of three cash crops, including soybean, corn, and canola, have been collected over Manitoba, Canada, during the Soil Moisture Active Validation Experimental 2012 (SMAPVEX-12) campaign. The results showed that GB and XGB have great potential in parameter estimation and remarkably improved accuracy. Our results also demonstrated a significant improvement in the dry biomass and LAI estimation compared to the previous studies. For LAI, the validation Root Mean Square Error (RMSE) was reported as 0.557 $m^2/m^2$ for canola using GB, and 0.298 $m^2/m^2$ for corn using GB, 0.233 $m^2/m^2$ for soybean using XGB. RMSE was reported for dry biomass as 26.29 $g/m^2$ for canola utilizing SVR, 57.97 $g/m^2$ for corn using RF, and 5.00 $g/m^2$ for soybean using GB. The results revealed that the deep artificial neural network had a better potential to estimate crop parameters than the ML algorithms.

**Keywords:** crop biomass; Leaf Area Index; Earth observations; Synthetic Aperture Radar; optical images; machine learning algorithms; SMAPVEX-12

## 1. Introduction

Due to rapid population growth and climate changes, global food security and agriculture production risks have been increased [1]. Information about annual crop production is vital for global and local food security. In particular, measuring and monitoring crop biophysical parameters, including dry biomass, crop height, crop density, and LAI, during the crop growing season are essential for improving crop growth models and yield estimation [1,2]. Biomass and LAI are two widely used crop parameters in crop monitoring and growth models [3,4]. As the input data in crop models, crop biophysical parameters are estimated using direct and indirect methods. The direct method consists of a ground measuring of the plant's parameters. These methods are usually destructive, costly, time-consuming, and complicated [5]. The information extracted from remote sensing data is non-destructive and significantly reduces time and cost. Remote sensing provides vital information on crop growth conditions over agricultural areas due to its extensive coverage and spatio-temporal resolution [6]. To this end, remote sensing imagery could be suitable for accurate crop monitoring.

Both SAR and optical data have been used to estimate crop parameters. Substantial studies have been carried out to investigate satellite optical data's potential to estimate various crop parameters. VIs extracted from optic bands are widely used to estimate crop parameters and monitor crop conditions. However, when the crop canopy is dense, optical data tend to be saturated [7]. In addition, since optical data in cloudy conditions are not helpful, SAR sensors use microwave wavelengths that can penetrate clouds and haze [1,8–13].

SAR sensors can also provide data in day and night without considering sun illumination, with suitable temporal coverage and sufficient spatial resolution [12,13]. Furthermore, soil and surface parameters and the crop canopy state can easily affect radar backscattering [14]. Moreover, the SAR backscattering coefficient is affected by crop and soil parameters. However, these effects have changed by various sensor parameters (i.e., wavelength, incidence angles, and polarization), different target parameters (i.e., canopy structure, water content, soil moisture, and soil roughness), and crop type and growth stage [1,11,15–18]. Thus, the combination of optical and SAR data has a great ability in crop monitoring.

Considerable researches have been conducted to estimate various crop parameters using satellite Earth observations, including RADARSAT-2 [1,17,19–21], RapidEye [5,19,20,22,23], Sentinel-1 [7,17,24–28], Sentinel-2 [7,25,29–32], Landsat-5 Thematic Mapper (TM) [33,34], Landsat-7 Enhanced Thematic Mapper Plus (ETM+) [34–36], Landsat-8 Operational Land Imager (OLI) [7,17,31,32,35–37], Worldview-2/3 [17,27,28,38,39], and MODIS [40,41].

The crop parameters estimation methods in the literature can be generally categorized into three groups: (1) parametric models, (2) non-parametric models, and (3) physically-based models [42]. Parametric models assume a clear relationship between input and output variables. In contrast, there is no assumption for the statistical distribution of input data in non-parametric models. Finally, physically-based models use physical laws, and model variables are frequently obtained from Radiative Transfer Models (RTMs) [42].

The new generation of satellite sensors coupled with an increasing need for big data mining has increased the essential need to use artificial intelligence (AI) for Earth observation data analysis. Machine learning (ML), a subset of AI, is learning algorithms by using training data. ML algorithms rapidly process a large amount of data and give helpful insight into the information leads to astonishing output. Another advantage of ML algorithms is that any apriori assumption is needed about data distribution [43]. Non-parametric MLAs, without any assumption for the statistical distribution of input data, have successfully been applied to remote sensing data to retrieve crop biophysical parameters, yield estimation, and crop mapping. Reisi Gahrouei, et al. [22] used an artificial neural network (ANN) and SVR to estimate LAI and dry biomass of three crops, including soybean, corn, and canola high-resolution RapidEye data. Reisi-Gahrouei, et al. [44] also used MLR and ANN to estimate crop biomass using UAVSAR data. Sharifi and Hosseingholizadeh [45] have investigated the potential of MLR, relevance vector regression (RVR), and SVR to estimate cereal height and biomass. Zhu, et al. [46] utilized unmanned

aerial vehicles (UAV) data to assess the ability of four MLAs, including MLR, RF, ANN, and SVR, to estimate the above-ground biomass (AGB). Luo, et al. [7] utilized MLR and SVR to estimate corn LAI and biomass. Deb, et al. [37] used parametric regression models and SVR to estimate agro-pastoral AGB. The excellent generalize capability of ML methods and their robustness to the noise in the case of low samples data makes them excellent tools to process remote sensing data and provide smart solutions in the field of precision agriculture.

In this study, we considered two scenarios to estimate crop parameters. First, we used four MLAs, including SVR, RF, GB, and XGB, in estimating two crop biophysical parameters. GB and XGB are the novel machine learning algorithms that received less attention in the crop parameters estimation method. This scenario was performed through three steps: (1) using polarimetric SAR features, (2) using optical VIs, and (3) using the integration of SAR and VIs features. These three steps could clearly show that the radar or optical remote sensing data or their combination in estimating crop parameters has excellent potential. Also, we used a deep artificial neural network to model the crop's LAI and dry biomass in the following scenario. In addition, the deep neural network received less attention in crop parameters estimation. Therefore, we tried to utilize the best performance from these algorithms using feature engineering and suitable parameters tuning. Several features counting VIs extracted from RapidEye spectral reflectance and polarimetric SAR decomposition feature extracted from UAVSAR data were selected to utilize as the input of mentioned algorithms. Moreover, the importance of each feature is investigated attentively. The results were compared, and the best method for estimating crop parameters was determined.

## 2. Materials and Methods

### 2.1. Study Area

The SMAPVEX12 campaign has been planned to support the calibration and validation of NASA's Soil Moisture Active Passive (SMAP) satellite mission products. This campaign was conducted over an agricultural area near Winnipeg, MB, Canada (Figure 1). The study area covers about 12.8 km by 70 km [47]. Five forested sites and 55 agriculture fields were selected for measuring dynamic and static crop (e.g., dry and wet biomass, LAI, crop height, crop density, etc.) and soil parameters (e.g., soil moisture, soil roughness, etc.), including 19 soybeans, eight corn and seven canola fields.

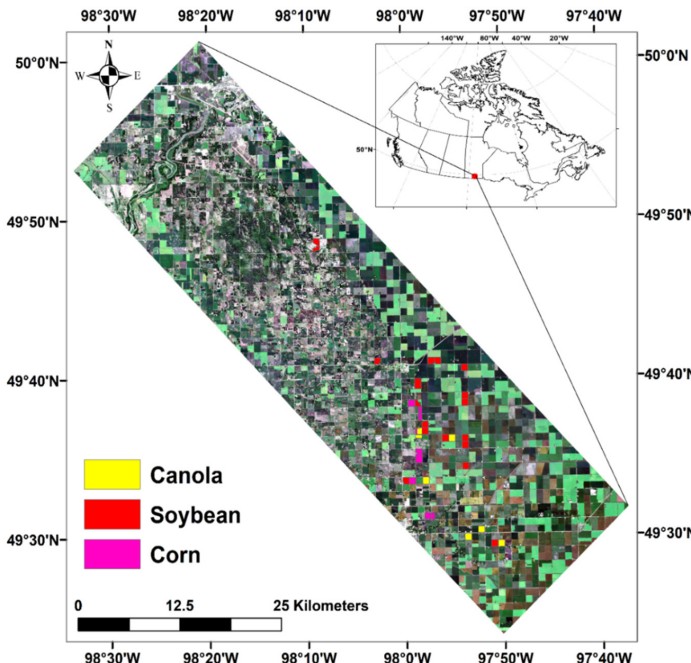

**Figure 1.** Location of study area ding SMAPVEX-12 campaign, the identified fields have been used in this research.

## 2.2. Sampling Strategy

In situ measurements from crop and soil parameters were collected over six weeks, from 7 June 2012 to 19 July 2012. The selected fields are relatively large, often 800 m × 800 m in size. In each field, 16 sampling sites were designed for sampling. However, crop parameters were measured only at three of these 16 sampling points. For canola, soybean, and corn, ground data collection was done through five plants along two rows (a total of ten plants). The wet biomass weight was immediately determined using a portable scale. Following wet weighing, wet biomass samples are placed in drying facilities for one week at 30 °C. After that, the weight of dry samples was determined. Crop's LAI was measured through the processing of hemispherical digital photos. Along two transacts in each field, seven photos were captured (a total of 14 photos). The CanEye software was then used to process these photos. The complete information about sampling strategy and collecting ground observations can be found in [47].

## 2.3. SAR Earth Observations

UAVSAR is a NASA L-band sensor that acquires SAR data in quad polarization mode. During SMAPVEX12, UAVSAR collected multitemporal high-resolution image data over the whole study area of the campaign. The detail of the UAVSAR sensor configurations and the acquisition date are shown in Table 1.

**Table 1.** UAVSAR sensor configuration and acquisition date.

| Data Characteristics | UAVSAR |
|---|---|
| Frequency | L-band (1.26 GHz) |
| Polarizations | Quad-Pol (HH, HV, VH, VV) |
| Spatial Resolution | 1.66 m range × 0.8 m azimuth SLC |
| Incident angle | 25 to 65 degrees (relevant: 35 to 45 degrees) |
| Acquisition dates | 17, 19, 22, 23, 25, 27, 29 June and 5, 8, 10, 13, 14, 17 July |

UAVSAR is a fully polarimetric SAR system that can collect a single-look complex (SLC) with 0.6 m × 1.6 m pixel spacing. Near to far range incidence angles are between 21.01° and 64.11°. UAVSAR is typically flown at 41,000 altitudes over the cross-track with a swath of approximately 20 km. The data were calibrated to complex cross-product format, multi-looked, and projected to ground range in simple geographic coordinates by NASA JPL and publicly available [44]. A 3 × 3 boxcar filter was applied to the data for speckle noise reduction. In addition to linear intensities (e.g., VV, HH, and HV), several polarimetric features were extracted from SAR data, including Cloude–Pottier decomposition components, i.e., entropy (H), anisotropy (A), and alfa angle ($\alpha$) [48,49], Freeman-Durden decomposition components [50], i.e., Surface ($O_F$), Double ($D_F$), and Volume ($V_F$) Scattering, and Yamaguchi decomposition components [51], Surface (OY), Double (DY), Volume (VY), and Helix (HY) Scattering. Besides, Radar Vegetation Index (RVI) was extracted and used as another model-based radar vegetation index using Equation (1):

$$\mathrm{RVI} = \frac{8\sigma^{\circ}_{HV}}{\sigma^{\circ}_{HH} + \sigma^{\circ}_{HV} + \sigma^{\circ}_{VV}} \tag{1}$$

## 2.4. Optical Earth Observations

Multi-temporal and multispectral RapidEye images were also used to estimate crop parameters. RapidEye constellation of five identical satellites acquires data in five bands, including blue, green, red, red-edge, and Near-Infrared. Each sensor's swath width is 77 km, and the ground sampling distance (GSD) at nadir is 6.5 m. The RapidEye images were atmospherically and geometrically corrected using PCI Geomatica's ATCOR2 and PCI Geomatica's OrthoEngine, respectively. The geometric correction was done using a rational function, satellite orbital information, ground control points (GCPs) collected from

Canada's National Road Network vector data, and the Canadian Digital Elevation Data (CDED). Several VIs were extracted from RapidEye images. The detail of the 11 VIs used in this study is shown in Table 2.

**Table 2.** Detail of the VIs extracted from RapidEye optic sensor used in this study.

| Index (Abbreviation) | Formula | Ref. |
| --- | --- | --- |
| Normalized Difference Vegetation Index (NDVI) | $\frac{R_{NIR} - R_R}{R_{NIR} + R_R}$ | [52] |
| Green NDVI (GNDVI) | $\frac{R_{NIR} - R_G}{R_{NIR} + R_G}$ | [53] |
| Simple Ratio (SR) | $\frac{R_{NIR}}{R_R}$ | [54] |
| Red-Edge Normalized Difference Vegetation Index (NDVI-RE) | $\frac{R_{NIR} - R_{RE}}{R_{NIR} + R_{RE}}$ | [55] |
| Red-Edge Simple Ratio(SR-RE) | $\frac{R_{NIR}}{R_{RE}}$ | [55] |
| Modified Triangular Vegetation Index (MTVI2) | $\frac{1.5\{1.2(R_{NIR} - R_G) - 2.5(R_R - R_G)\}}{\sqrt{(2R_{NIR} + 1)^2 - (6R_{NIR} - 5\sqrt{R_R}) - 0.5}}$ | [56] |
| Enhanced Vegetation Index (EVI) | $\frac{2.5(R_{NIR} - R_R)}{1 + R_{NIR} + 6R_R - 7.5R_B}$ | [57] |
| Green Chlorophyll Index (CL-G) | $R_{NIR} - R_G - 1$ | [58–60] |
| Red-Edge Chlorophyll Index (CL-RE) | $R_{NIR} - R_{RE} - 1$ | [58–60] |
| MERIS Trrestrial Chlorophyll Index (MTCI) | $\frac{R_{NIR} - R_{RE}}{R_R + R_{RE}}$ | [61] |
| Soil-Adjusted Vegetation Index (SAVI) | $\frac{(1 + 0.5)(R_{NIR} - R_R)}{R_{NIR} + R_R + 0.5}$ | [62] |

## 3. Methodology

MLAs are recently used in classification and regression problems in many areas. In this study, regression models, e.g., RF, GB, XGB, and SVR, were used to estimate crop's LAI and dry biomass. MLAs were implemented using the open-source Python Scikit-learn package. Besides, deep ANN was implemented using the Keras package. The data were divided into train and test data. Two-thirds (i.e., ~66.7%) of the data were selected to train the models, and the remaining data (i.e., ~33.3%) were used as test data. In this study, first, we calculate the correlation between SAR and optical VIs.

Then, RF feature importance was calculated for each crop and crop parameter separately. The less important features were removed by considering the absolute value of feature importance greater than 0.9 between features. Finally, the remaining features were combined to estimate crop parameters. The selected SAR and VIs features were separately fed into the model as the input data in the first and second steps. Then, the combining of SAR and VIs features was used for modeling. The results of the three separate input data were compared to each other. Furthermore, a deep artificial neural network based on the selected feature was designed and implemented. Furthermore, Grid Search Cross-Validation (GridSearchCV) was used to tune the hyper-parameters of all the ML algorithms.

### 3.1. Random Forest Regression

RF is a robust ensemble learning method, which is widely used in classification and regression problems. Ensemble learning is the process in which multiple models are produced and combined to solve a particular task. Two common types of ensemble learning are boosting and bagging. Bagging is made up of fitting several models that train independently to reduce variance to avoid overfitting while improving combined models' stability and accuracy [63]. RF is a successful bagging approach made up of a substantial number of individual decision trees. Each tree makes its prediction. Finally,

the model combines all predictions to obtain a better performance [64]. Each tree grows independently using a bootstrap sampling of the training data [29].

In contrast to the linear regression model, an RF regressor model cannot predict outlier data, e,g, predicting the data from training samples. Various researches have used RF regression and classification models to estimate crop parameters or map croplands [31,40,46,65]. The GridSearchCV parameters used in the RF are shown in Table 3.

**Table 3.** Grid Search parameters used in the RF model.

| Parameters | Description | Grid Search Values |
|---|---|---|
| n_estimators | No. of trees in the forest | 5, 10, 25, 50, 100 |
| max_depth | Maximum depth of the trees | 2, 3, 5, 8, 10 |
| min_samples_split | Minimum number of samples required to split an internal node | 2, 3, 5, 10 |
| min_samples_leaf | Minimum number of samples required to be at a leaf node | 1, 2, 3, 5, 10 |

### 3.2. Support Vector Regression

The support vector machines (SVMs) algorithm, developed by Vapnik and his colleagues [66], is one of the most widely used kernel-based machine learning algorithms, which is used in a variety of problems, especially in classification tasks [63]. Maintaining all the algorithm's main features, like maximal margin, SVM can also be used in regression problems. SVR, firstly introduced by Drucker, et al. [67], has several minor differences from SVM. The regression model's output has infinite numbers, but in SVM, the output is finite numbers.

In regression models, a margin of tolerance (epsilon) is set in approximation. There will be various reasons that make regression models more complicated than the SVM model. SVR gives us the flexibility to define how much error is acceptable in our model and find an appropriate line (or hyperplane in higher dimensions) to fit the data. In this manner, the tube's points, the points outside the tube, receive penalization; however, the prediction function receives no penalization either above or below. SVR and SVM are widely used in recent researches to estimate crop parameters and cropland mapping [7,26,31,37,45,46,65,68,69]. The Grid Search parameters used for the SVR model are shown in Table 4.

**Table 4.** GridSearchCV parameters for SVR.

| Parameters | Description | Grid Search Values |
|---|---|---|
| Kernel | Specifies the kernel type to be used in the algorithm | 'linear', 'rbf', 'poly' |
| Gamma | Kernel coefficient for 'rbf', 'poly' and 'sigmoid' | 0.001, 0.01, 0.1, 0.5, 0.8, 1, 3 |
| Degree | degree of polynomial | 2, 3 |
| C | Penalty parameter | 5, 10, 20, 50, 100, 200, 500, 1000 |

### 3.3. Gradient Boosting and Extreme Gradient Boosting

GB regression algorithms were subsequently developed by Friedman [70,71]. As we said in part2, two common types of ensemble learning are boosting and bagging. GB, a machine learning method, is an extension of the boosting method. GB, like RF, is used in regression and classification tasks. GB method is based on minimizing a loss function, and various types of loss functions can be used. The regularization techniques are customarily used to reduce overfitting effects. GB negligibly has been used in crop biomass estimation [31,65]. One of the most attractive gradients boosting implementations is XGB [72],

first started by Tianqi Chen (Tianqi Chen on http://datascience.la/xgboost-workshop-and-meetup-talk-with-tianqi-chen/, accessed on 3 July 2021) as a research project. It is an ensemble machine learning algorithm that uses a gradient boosting framework. XGB is designed to enhance a machine learning model's performance, speed, flexibility, and efficiency. The Grid Search parameters used for the GB and XGB algorithms are shown in Table 5.

**Table 5.** GridSearchCV parameters for the GB and XGB algorithms.

| Parameters | Description | Grid Search Values |
|---|---|---|
| learning_rate | Shrinks the contribution of each tree | 0.001, 0.005, 0.01, 0.05, 0.1, 0.15, 0.3 |
| n_estimators | The number of boosting stages to conduct. | 10, 25, 50, 70, 100 |
| max_depth | Limits the number of nodes in the tree. | 2, 3, 5, 7, 10 |
| max_features | The number of features to consider when searching for the best split. | 'auto', 'sqrt', 'log2' |

### 3.4. Deep Artificial Neural Network Regression

ANNs are popular machine learning algorithms inspired by the human brain [64]. A simplified model of the brain shows a considerable number of primary computing devices called neurons. Through these substantially connected neurons, highly complex computations can be carried out. ANN consists of interconnected neurons that learn by adopting and modifying the weights [29]. This model typically includes one input layer, more than two hidden layers, and one output layer. In the ANN model, neurons of one layer can be connected to all other layers' neurons but not to the same layer's neurons. Each neuron is connected to all neurons in the previous and following layers in a fully connected ANN [73].

In this study, we used a dense, deep ANN. The primary considerations for tuning hyper-parameters of ANN are the number of neurons and hidden layers. Several empirical methods can determine the number of neurons in each layer [74,75]. In this study, we have determined the number of the neurons using Equation (2) [74]:

$$N_n = \sqrt{(m+2)N} + 2\sqrt{\frac{N}{m+2}} \tag{2}$$

In this equation, $N_n$ is the number of neurons in each layer, $N$ is the number of input neurons, and $m$ is the number of layers. We examined various activation functions for the deep ANN model, including ReLU, Tanh, Sigmoid, and Linear. Adam's optimization method, an extension of Stochastic Gradient Descent (SGD), was used to update the network's weight iteratively. The early stopping approach was used to avoid overfitting. Furthermore, 20% of the training sample was selected as the validation data.

### 3.5. Evaluation Criteria

Several criteria were used to evaluate prediction performance, including RMSE, mean absolute error (MAE), and Pearson correlation coefficient (R). The formula of the RMSE is as follows:

$$RMSE = \sqrt{\frac{\sum_{i=1}^{N}(\hat{y}_i - y_i)^2}{N}} \tag{3}$$

where $N$ is the number of data.

In addition, the normalized RMSE (nRMSE) ($\frac{\text{RMSE}}{\text{related crop parameter' average}}$) is presented in one figure for better and accurate visualization. MAE is calculated as the following equation:

$$\text{MAE} = \frac{\sum_{i=1}^{N}|\hat{y}_i - y_i|}{N} \tag{4}$$

R is used in statistics problems to measure how the relationship between predicted and observed data is robust:

$$\text{R} = \frac{\left(N\sum_{i=1}^{N}(\hat{y}_i y_i)\right) - \left(\sum_{i=1}^{N}\hat{y}_i\right)\left(\sum_{i=1}^{N}y_i\right)}{\sqrt{\left(N\sum_{i=1}^{N}\hat{y}_i{}^2 - \left(\sum_{i=1}^{N}\hat{y}_i\right)^2\right)\left(N\sum_{i=1}^{N}y_i{}^2 - \left(\sum_{i=1}^{N}y_i\right)^2\right)}} \tag{5}$$

## 4. Results and Discussion

Several optical VIs and polarimetric SAR data were extracted from RapidEye and UAVSAR data to explore satellite data's potential to evaluate and estimate crop parameters. The results showed acceptable agreement with the researches had done before by Hosseini, et al. [9] and Reisi Gahrouei, et al. [22]. The impact of optical VIs, UAVSAR polarimetric features, and integrating them on the accuracy of retrieving dry biomass and LAI using four machine learning regression models is assessed in the following sections.

### 4.1. Time Series Analysis of Radar Backscattering

Figure 2 presents the temporal profiles of the three crops. The left axes represent three SAR backscattering, including VV, HH, and HV. The right axes in the left images are regarding dry biomass, while the right images are regarding LAI. For canola, all three intensities, including VV, HH, and HV, generally decrease from 17 June to 14 July 2012. As expected, the amount of dry biomass during the campaign increased, while the LAI reduced from start to middle of the campaign. Generally, all three SAR backscattering coefficients from 17 June to 14 July 2012, are rising for corn. Also, the amount of dry biomass and LAI increased during the campaign. For soybean, in total, the HV and HH backscattering coefficient is rising, but the VV behavior is irregular. For soybean, similar to corn, the amount of dry biomass and LAI increased during the campaign.

### 4.2. Correlation and Features Importance

Correlation coefficients between all features extracted for each crop have shown in Figure 3. For canola, the correlation between DF and DY, OF and OY, and VF and VY is high. The absolute correlation between DF and most of the other radar features is generally higher than 0.9. Overall, the correlation between OF and OY with other decompositions is relatively low. The correlation between DF and DY with the other radar features is also high. As well, the correlation between radar features with VIs is low. Between VIs, approximately in most cases, correlation is high. Apart from A, H, and Alpha, the correlation between other SAR features is relatively high for corn. However, the number of radar features with an absolute correlation exceeded 0.9 is negligible. The correlation between $D_F$ and $D_Y$, $O_F$ and $O_Y$, and $V_F$ and $V_Y$ is higher than 0.9. Besides, a high correlation can be seen between VIs. Compared to corn and canola, the correlation between SAR features is relatively low.

For corn's dry biomass, the higher importance is related to MCTI VI. However, between 5 high important features, four of them are SAR parametric features. For corn LAI, similar to dry corn biomass, the higher feature importance is MCTI. Nevertheless, in contrast to corn's dry biomass, from 5 higher importance features, four of them are related to spectral VI. For canola's dry biomass, the higher importance is related to $D_F$.

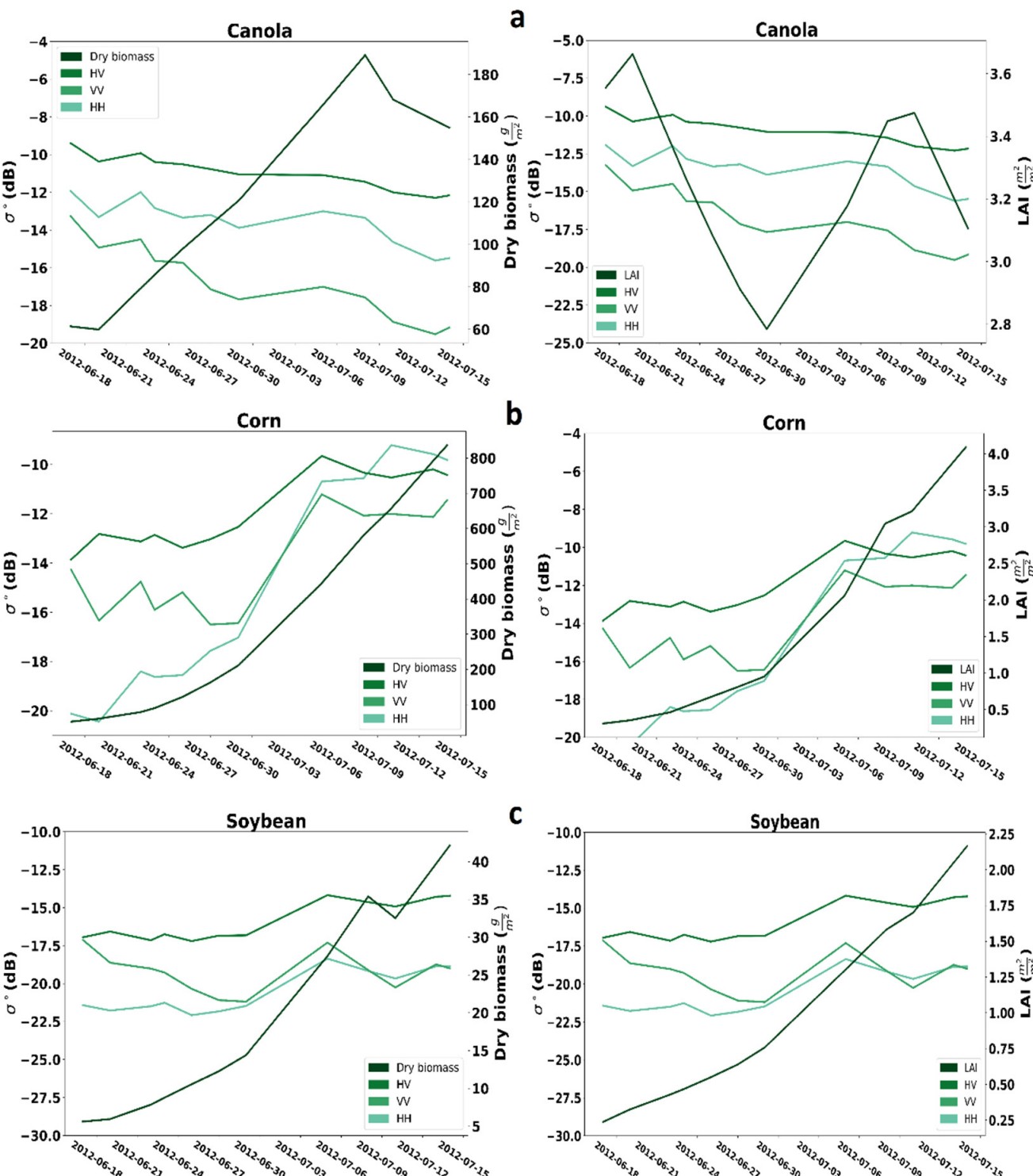

**Figure 2.** Temporal profile analysis for SAR backscatter coefficient of dry biomass and LAI for (**a**) canola, (**b**) corn, (**c**) soybean.

Of five higher importances, three are related to SAR parameters, and the remaining are regarding spectral VIs. For canola LAI, the higher importance is related to RVI. Also, between five high importance features, four of them are regarding SAR parameters. For soybean's dry biomass, the higher importance is related to CL-EDGE. Besides, four out of five higher importance are related to spectral VIs. Finally, for soybean LAI, similar to

soybean's dry biomass, CL-EDGE has higher importance. In addition, between 5 higher importance, four of them are related to spectral VIs.

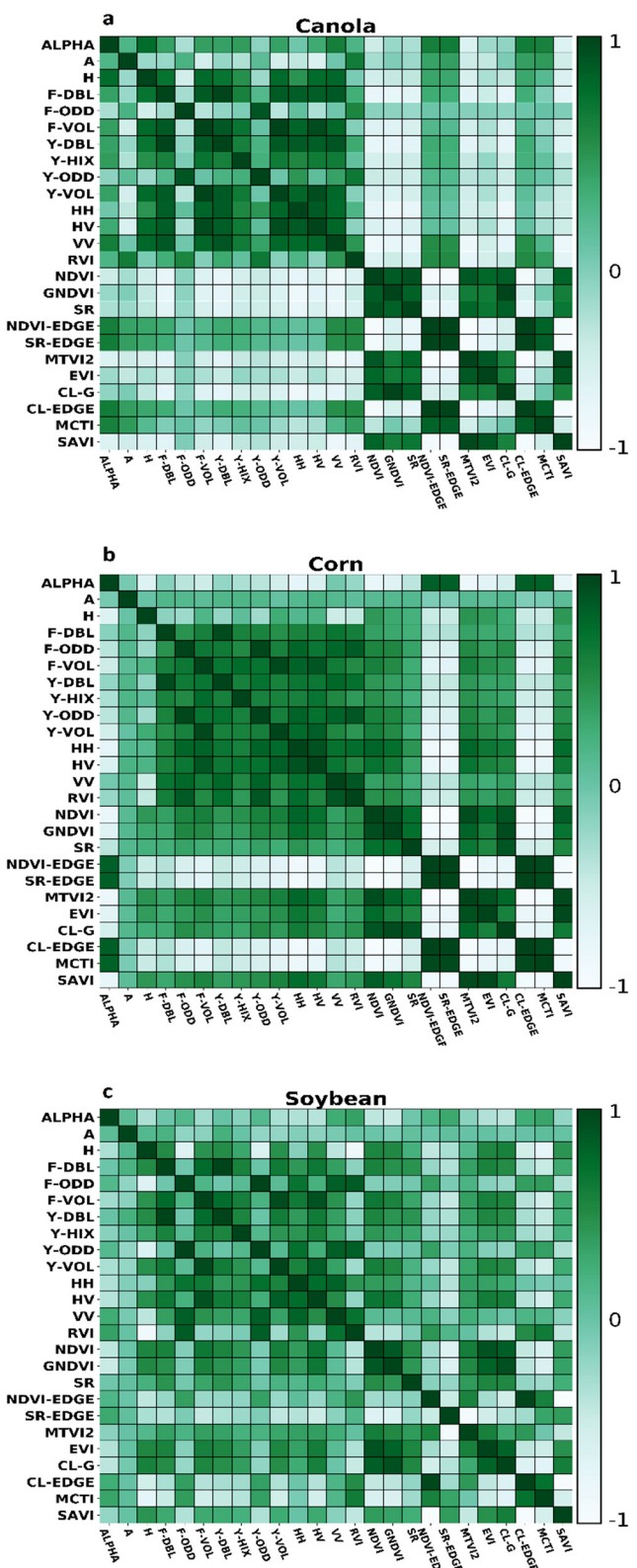

**Figure 3.** Correlation analysis calculated for SAR and optic features for (**a**) Canola, (**b**) Corn, (**c**) Soybean.

The details of the RF feature importance are listed in Table 6. The color of the feature cell with higher importance tends to be green, and the low important feature's color tend to be yellow. Details of the features used in each crop parameter can be seen in Table 7.

**Table 6.** RF feature importance results.

|  | Dry Biomass | | | LAI | | |
|---|---|---|---|---|---|---|
|  | **Corn** | **Canola** | **Soybean** | **Corn** | **Canola** | **Soybean** |
| HH | 0.235 | 0.032 | 0.024 | 0.191 | 0.013 | 0.006 |
| HV | 0.006 | 0.009 | 0.005 | 0.038 | 0.005 | 0.004 |
| VV | 0.004 | 0.009 | 0.005 | 0.002 | 0.355 | 0.004 |
| ALPHA | 0.020 | 0.088 | 0.037 | 0.038 | 0.230 | 0.012 |
| A | 0.011 | 0.052 | 0.015 | 0.007 | 0.011 | 0.005 |
| H | 0.009 | 0.057 | 0.008 | 0.007 | 0.019 | 0.025 |
| F_ODD | 0.012 | 0.028 | 0.007 | 0.002 | 0.017 | 0.010 |
| F_DBL | 0.007 | 0.086 | 0.005 | 0.007 | 0.084 | 0.005 |
| F_VOL | 0.041 | 0.011 | 0.004 | 0.043 | 0.008 | 0.007 |
| Y_ODD | 0.003 | 0.008 | 0.003 | 0.002 | 0.009 | 0.007 |
| Y_DBL | 0.002 | 0.006 | 0.012 | 0.003 | 0.015 | 0.007 |
| Y_VO | 0.015 | 0.022 | 0.021 | 0.035 | 0.013 | 0.013 |
| Y_HIX | 0.010 | 0.019 | 0.011 | 0.013 | 0.013 | 0.004 |
| RVI | 0.009 | 0.057 | 0.007 | 0.004 | 0.021 | 0.008 |
| NDVI | 0.009 | 0.003 | 0.092 | 0.025 | 0.013 | 0.083 |
| GNDVI | 0.020 | 0.005 | 0.043 | 0.009 | 0.020 | 0.009 |
| SR | 0.005 | 0.018 | 0.017 | 0.058 | 0.001 | 0.007 |
| NDVI_EDGE | 0.005 | 0.102 | 0.218 | 0.051 | 0.005 | 0.288 |
| SR_EDGE | 0.010 | 0.103 | 0.012 | 0.034 | 0.007 | 0.038 |
| MTVI2 | 0.007 | 0.026 | 0.011 | 0.017 | 0.026 | 0.005 |
| EVI | 0.005 | 0.005 | 0.011 | 0.006 | 0.015 | 0.007 |
| CL_G | 0.004 | 0.005 | 0.035 | 0.008 | 0.002 | 0.009 |
| CL_EDGE | 0.018 | 0.127 | 0.296 | 0.096 | 0.033 | 0.409 |
| MCTI | 0.523 | 0.081 | 0.085 | 0.289 | 0.010 | 0.024 |
| SAVI | 0.010 | 0.040 | 0.017 | 0.017 | 0.004 | 0.004 |

**Table 7.** SAR and Optical features used for different crop parameter modeling.

|  | Dry Biomass | | | LAI | | |
|---|---|---|---|---|---|---|
|  | **Corn** | **Canola** | **Soybean** | **Corn** | **Canola** | **Soybean** |
| HH | ✓ | ✓ | ✓ | ✓ | ✓ | ✓ |
| HV | ✗ | ✓ | ✓ | ✗ | ✓ | ✓ |
| VV | ✓ | ✓ | ✓ | ✓ | ✓ | ✓ |
| ALPHA | ✓ | ✓ | ✓ | ✓ | ✓ | ✓ |
| A | ✓ | ✓ | ✓ | ✓ | ✓ | ✓ |

**Table 7.** *Cont.*

| | Dry Biomass | | | LAI | | |
|---|---|---|---|---|---|---|
| | Corn | Canola | Soybean | Corn | Canola | Soybean |
| H | ✓ | ✓ | ✓ | ✓ | ✓ | ✓ |
| F_ODD | ✓ | ✗ | ✗ | ✓ | ✗ | ✗ |
| F_DBL | ✗ | ✗ | ✗ | ✗ | ✗ | ✗ |
| F_VOL | ✓ | ✗ | ✗ | ✓ | ✗ | ✗ |
| Y_ODD | ✗ | ✓ | ✓ | ✗ | ✓ | ✓ |
| Y_DBL | ✓ | ✓ | ✓ | ✓ | ✓ | ✓ |
| Y_VO | ✗ | ✓ | ✓ | ✗ | ✓ | ✓ |
| Y_HIX | ✓ | ✓ | ✓ | ✓ | ✓ | ✓ |
| RVI | ✓ | ✓ | ✓ | ✓ | ✓ | ✓ |
| NDVI | ✗ | ✓ | ✓ | ✓ | ✓ | ✓ |
| GNDVI | ✗ | ✗ | ✗ | ✗ | ✗ | ✗ |
| SR | ✓ | ✓ | ✓ | ✓ | ✓ | ✓ |
| NDVI_EDGE | ✗ | ✗ | ✗ | ✗ | ✗ | ✗ |
| SR_EDGE | ✗ | ✓ | ✓ | ✗ | ✓ | ✓ |
| MTVI2 | ✗ | ✓ | ✓ | ✗ | ✓ | ✓ |
| EVI | ✗ | ✗ | ✗ | ✓ | ✗ | ✗ |
| CL_G | ✓ | ✓ | ✓ | ✓ | ✓ | ✓ |
| CL_EDGE | ✗ | ✓ | ✓ | ✗ | ✓ | ✓ |
| MCTI | ✓ | ✓ | ✓ | ✓ | ✓ | ✓ |
| SAVI | ✓ | ✗ | ✗ | ✗ | ✗ | ✗ |

*4.3. Sensitivity Analysis*

Complete information over validation and calibration accuracies for retrieving dry biomass and LAI for each crop is shown in Tables 8 and 9, respectively. The following details are for validation data. As maturity methods, canola builds up appreciable plant material and vegetation water. This considerable water volume might lead to a greater tendency towards saturation of signals from canola canopies, especially for SAR backscatter. For canola' dry biomass, generally, the accuracy of integrated input data was better than the SAR parameters or spectral VIs, separately. The prediction performance of optical VIs was slightly better than the SAR features using. The best performance was related to SVR MLA with RMSE = 26.29 $g/m^2$, MAE = 20.72 $g/m^2$, and R = 0.95 using a combination of SAR and optical data (Figure 4a). A low amount of overestimated value can be seen in the early growth stage. Moreover, SVR underestimated dry biomass for canola at advanced development stages. Low error among dry biomass estimation using VIs spectral data provided by GB method with RMSE of 38.87 g/m2 and MAE of 28.34 g/m2, considerably higher than integrated estimation error. Using SAR polarimetric features, high accuracy was delivered by RF algorithms (RMSE = 46.67 $g/m^2$, 33.45 $g/m^2$, and R = 0.83). The results showed no saturation in the high and low amount of dry biomass prediction using a combination of optic and SAR data.

**Table 8.** Results of four MLAs in estimating dry biomass.

| | | | Dry Biomass | | | | | | | | |
| | | | Canola | | | Corn | | | Soybean | | |
| | | | R | RMSE | MAE | R | RMSE | MAE | R | RMSE | MAE |
|---|---|---|---|---|---|---|---|---|---|---|---|
| RF | Calibration | Optical | 0.94 | 29.49 | 21.33 | 0.92 | 78.68 | 49.87 | 0.92 | 6.18 | 4.25 |
| | | SAR | 0.92 | 33.137 | 23.58 | 0.92 | 94.25 | 58.6 | 0.94 | 5.05 | 3.75 |
| | | Integrated | 0.97 | 24.19 | 18.14 | 0.96 | 63.73 | 39.15 | 0.96 | 4.35 | 3.15 |
| | Validation | Optical | 0.85 | 43.35 | 35.56 | 0.94 | 77.5 | 49.19 | 0.88 | 7.07 | 4.69 |
| | | SAR | 0.83 | 46.67 | 33.45 | 0.91 | 98.04 | 60.83 | 0.9 | 6.79 | 4.87 |
| | | Integrated | 0.91 | 36.74 | 30.02 | 0.96 | 57.97 | 41.15 | 0.94 | 5.5 | 3.7 |
| SVR | Calibration | Optical | 0.93 | 31.26 | 18.55 | 0.94 | 72.53 | 42.56 | 0.85 | 8.01 | 5.04 |
| | | SAR | 0.91 | 34.54 | 18.61 | 0.9 | 96.84 | 60.89 | 0.87 | 7.73 | 5.02 |
| | | Integrated | 0.97 | 21.36 | 10.78 | 0.94 | 76.53 | 47.68 | 0.96 | 4.4 | 2.18 |
| | Validation | Optical | 0.86 | 42.03 | 32.53 | 0.91 | 76.55 | 51.75 | 0.85 | 7.76 | 5.2 |
| | | SAR | 0.8 | 49.46 | 34.46 | 0.95 | 60.2 | 46.94 | 0.89 | 6.88 | 4.43 |
| | | Integrated | 0.95 | 26.29 | 20.72 | 0.96 | 58.62 | 42.34 | 0.92 | 5.79 | 3.75 |
| GB | Calibration | Optical | 0.93 | 28.61 | 18.31 | 0.94 | 71.28 | 44.97 | 0.92 | 6.12 | 4.17 |
| | | SAR | 0.91 | 34.14 | 18.25 | 0.87 | 93.58 | 61.2 | 0.93 | 5.4 | 3.81 |
| | | Integrated | 0.95 | 24.61 | 11.84 | 0.94 | 75.25 | 45.23 | 0.96 | 3.56 | 2.19 |
| | Validation | Optical | 0.88 | 38.87 | 28.34 | 0.94 | 69.85 | 44.37 | 0.79 | 8.64 | 6.13 |
| | | SAR | 0.81 | 47.86 | 32.19 | 0.92 | 96.4 | 63.95 | 0.86 | 7.22 | 4.87 |
| | | Integrated | 0.93 | 30.77 | 24.3 | 0.96 | 63.02 | 40.28 | 0.94 | 5 | 3.5 |
| XGB | Calibration | Optical | 0.91 | 30.25 | 19.56 | 0.94 | 72.38 | 45.76 | 0.94 | 5.22 | 3.46 |
| | | SAR | 0.91 | 34.84 | 18.49 | 0.89 | 91.15 | 61.68 | 0.93 | 5.6 | 3.97 |
| | | Integrated | 0.95 | 24.687 | 12.15 | 0.94 | 72.02 | 45.01 | 0.97 | 3.67 | 2.41 |
| | Validation | Optical | 0.86 | 40.53 | 30.38 | 0.93 | 77.87 | 48.03 | 0.84 | 7.41 | 4.93 |
| | | SAR | 0.79 | 50.26 | 33.62 | 0.91 | 92.14 | 62.39 | 0.89 | 6.98 | 4.99 |
| | | Integrated | 0.93 | 31.79 | 23.18 | 0.95 | 68.73 | 42.1 | 0.94 | 5.25 | 3.58 |

**Table 9.** Results of four MLAs in estimating LAI.

| | | | LAI | | | | | | | | |
| | | | Canola | | | Corn | | | Soybean | | |
| | | | R | RMSE | MAE | R | RMSE | MAE | R | RMSE | MAE |
|---|---|---|---|---|---|---|---|---|---|---|---|
| RF | Calibration | Optical | 0.95 | 0.602 | 0.4 | 0.92 | 0.396 | 0.252 | 0.89 | 0.351 | 0.258 |
| | | SAR | 0.89 | 0.856 | 0.55 | 0.94 | 0.339 | 0.238 | 0.95 | 0.236 | 0.18 |
| | | Integrated | 0.96 | 0.555 | 0.368 | 0.96 | 0.302 | 0.203 | 0.97 | 0.174 | 0.115 |
| | Validation | Optical | 0.92 | 0.771 | 0.577 | 0.9 | 0.442 | 0.287 | 0.83 | 0.368 | 0.262 |
| | | SAR | 0.71 | 1.204 | 0.798 | 0.94 | 0.355 | 0.253 | 0.9 | 0.292 | 0.218 |
| | | Integrated | 0.93 | 0.699 | 0.515 | 0.95 | 0.321 | 0.204 | 0.93 | 0.239 | 0.167 |
| SVR | Calibration | Optical | 0.97 | 0.432 | 0.276 | 0.95 | 0.414 | 0.249 | 0.88 | 0.345 | 0.228 |
| | | SAR | 0.88 | 0.891 | 0.565 | 0.94 | 0.358 | 0.206 | 0.92 | 0.295 | 0.209 |
| | | Integrated | 0.97 | 0.371 | 0.241 | 0.96 | 0.306 | 0.179 | 0.96 | 0.194 | 0.129 |
| | Validation | Optical | 0.93 | 0.674 | 0.562 | 0.89 | 0.454 | 0.283 | 0.84 | 0.364 | 0.254 |
| | | SAR | 0.78 | 1.244 | 0.797 | 0.91 | 0.399 | 0.234 | 0.9 | 0.291 | 0.209 |
| | | Integrated | 0.94 | 0.579 | 0.459 | 0.95 | 0.316 | 0.224 | 0.92 | 0.261 | 0.183 |

**Table 9.** *Cont.*

| | | | LAI | | | | | | | | |
| | | | Canola | | | Corn | | | Soybean | | |
| | | | R | RMSE | MAE | R | RMSE | MAE | R | RMSE | MAE |
|---|---|---|---|---|---|---|---|---|---|---|---|
| GB | Calibration | Optical | 0.96 | 0.451 | 0.292 | 0.93 | 0.457 | 0.286 | 0.92 | 0.279 | 0.199 |
| | | SAR | 0.89 | 0.83 | 0.464 | 0.93 | 0.361 | 0.212 | 0.94 | 0.236 | 0.167 |
| | | Integrated | 0.95 | 0.564 | 0.25 | 0.95 | 0.329 | 0.219 | 0.96 | 0.196 | 0.131 |
| | Validation | Optical | 0.93 | 0.645 | 0.542 | 0.92 | 0.424 | 0.302 | 0.84 | 0.384 | 0.283 |
| | | SAR | 0.78 | 1.082 | 0.641 | 0.94 | 0.371 | 0.266 | 0.89 | 0.322 | 0.222 |
| | | Integrated | 0.95 | 0.557 | 0.399 | 0.96 | 0.298 | 0.219 | 0.94 | 0.233 | 0.167 |
| XGB | Calibration | Optical | 0.97 | 0.405 | 0.273 | 0.93 | 0.38 | 0.268 | 0.9 | 0.318 | 0.222 |
| | | SAR | 0.89 | 0.812 | 0.456 | 0.97 | 0.251 | 0.173 | 0.94 | 0.268 | 0.202 |
| | | Integrated | 0.92 | 0.591 | 0.343 | 0.95 | 0.31 | 0.208 | 0.98 | 0.133 | 0.055 |
| | Validation | Optical | 0.93 | 0.646 | 0.527 | 0.92 | 0.399 | 0.273 | 0.84 | 0.36 | 0.247 |
| | | SAR | 0.78 | 1.093 | 0.645 | 0.95 | 0.321 | 0.224 | 0.9 | 0.293 | 0.213 |
| | | Integrated | 0.93 | 0.635 | 0.499 | 0.95 | 0.315 | 0.222 | 0.94 | 0.233 | 0.164 |

LAI is indicative of the crop structure and affects both reflectance and backscatter at canopy scales. For canola LAI, the integration of SAR polarimetric data and spectral VIs in GB and XGB improved performance. However, optical VIs outperforms the integration of SAR and optical features in SVR and RF. Besides, the accuracy of LAI estimation with VIs spectral data was better than SAR polarimetric data. As demonstrated by Figure 4b, LAI for canola estimated by GB using both SAR and VIs feature was highly correlated with in situ measured LAI (0.557 $m^2/m^2$, 0.399 $m^2/m^2$, and R = 0.95). However, GB slightly underestimates the LAI in the mid-growth stage. The higher accuracy with spectral VIs regarded to GB and XGB, with approximately the same RMSE, but better MAE in XGB. A minimal amount of saturation could be seen in high values of LAI.

For corn's dry biomass, like canola's dry biomass, integrated features have higher accuracy. The performance of optical VIs data was better than SAR polarimetric parameters in RF, GB, and XGB, while in SVR, the accuracy of SAR polarimetric features is higher than VIs spectral data. The prediction accuracy delivered by RF, SVR, and GB has the same R, but RF has a lower error than SVR and GB. The higher accuracy was related to the RF regression model with RMSE = 57.97 $g/m^2$, MAE = 41.15 $g/m^2$, and R = 0.96 (Figure 4c). Although a few ground measurements are available at high dry biomass, observed and estimated, dry biomass values are well distributed about the 1:1 line. The best performance of optical VIs regarded to GB regression model with RMSE = 69.85 $g/m^2$, MAE = 44.37 $g/m^2$, R = 0.94. The higher accuracy with SAR polarimetric data was related to the SVR model with RMSE = 60.2 $g/m^2$, MAE = 46.94 $g/m^2$, and R = 0.94. The result of RF show low saturation in the high amount of corn's dry biomass.

For corn's LAI, the best performance was related to integrating SAR polarimetric data and spectral VIs. Besides, the estimation accuracy of spectral VIs is slightly worse than SAR polarimetric data. The best accuracy was regarding the GB regression method with RMSE = 0.298 $m^2/m^2$, MAE = 0.219 $m^2/m^2$, and R = 0.96 using a combination of SAR and optical VIs features (Figure 4d). Early in the season, the GB overestimates LAI. This may be due to a more open canopy in the early growth stages, leaving more soil exposed to soil properties, contributing significantly to reflectance and backscatter. The RMSE of SVR, RF, and XGB is nearly equal. The best performance for spectral VIs was provided by the XGB model with RMSE = 0.399 $m^2/m^2$, MAE = 0.273 $m^2/m^2$, and R = 0.92. The best performance for SAR data was delivered by the XGB model with RMSE = 0.321 $m^2/m^2$, MAE = 0.221 $m^2/m^2$, and R = 0.95.

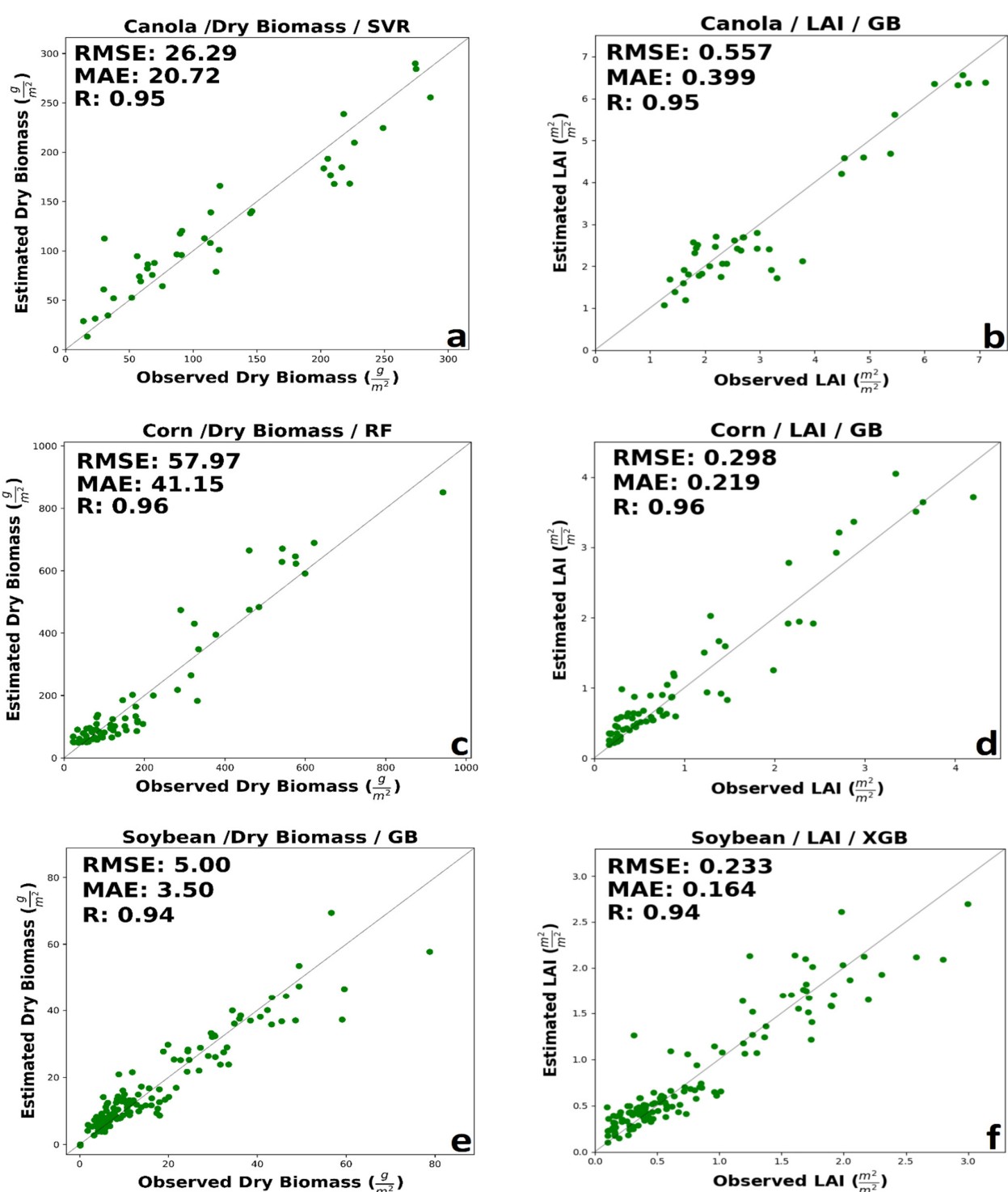

**Figure 4.** The best performance of machine learning regression models in estimating (**a**) canola dry biomass, (**b**) canola LAI, (**c**) corn dry biomass, (**d**) corn LAI, (**e**) soybean dry biomass, and (**f**) soybean LAI.

For soybean's dry biomass, generally, the integration of SAR and optical data had better performance. The estimation performance of SAR polarimetric data was slightly better than optic data in all cases. The best performance among MLAs was related to GB regression model with RMSE = 5.00 g/m$^2$, MAE = 3.5 g/m$^2$, and R = 0.94 (Figure 4e). SVR had the lower MAE among all algorithms; however, RF has the lower RMSE comparison to SVR. Also, RF and GB had a similar MAE, but RF had the lower RMSE. ML algorithms had a significant saturation in the high value of dry biomass (higher than 60 g/m$^2$).

For soybean's LAI, like corn and canola, the best performance belongs to integrating SAR and optic data. Also, the accuracy of SAR polarimetric data, compared to spectral VIs, was better. The XGB and GB had the same RMSE, however, XGB had a better MAE (RMSE = 0.233 m$^2$/m$^2$, MAE = 0.164 m$^2$/m$^2$, and R = 0.94 (Figure 4f). The higher accuracy using SAR data was related to SVR with RMSE = 0.291 m$^2$/m$^2$, MAE = 0.209 m$^2$/m$^2$, and R = 0.9. Using optical VIs data XGB provided better results with RMSE = 0.36 m$^2$/m$^2$, MAE = 0.247 m$^2$/m$^2$, and R = 0.84. The best performance in estimating dry biomass and LAI for each crop is shown in Figure 4.

Figure 5 shows the results of four MLAs' nRMSE for three crops shown in the boxplot. The quartile of distribution shows in the box. The rest of the dataset showed whiskers. Each boxplot's data consisted of each method's calibration and validation data, including three various input data (SAR polarimetric features, VIs spectral data, and integration of SAR and optical features). The results showed that MLAs performed better in canola rather than corn and soybean. For corn's dry biomass, SVR has better accuracy rather than the other methods. In addition, the median for XGB in corn dry biomass and LAI are lower, which means better accuracy.

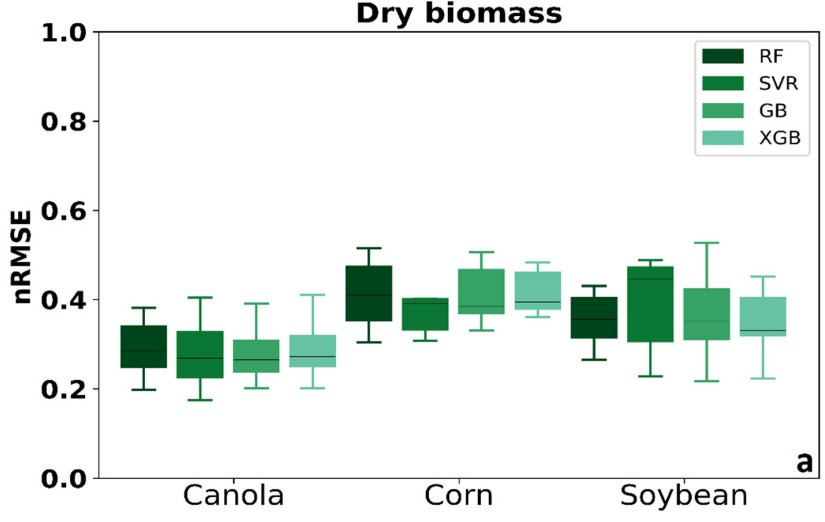

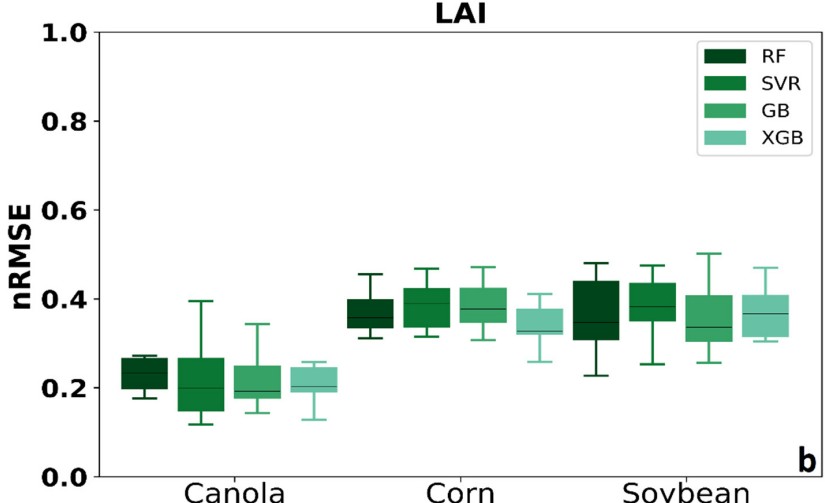

**Figure 5.** Comparing four MLAs' nRMSE for three crops (**a**) dry biomass and (**b**) LAI.

*4.4. The Results of Deep Neural Network*

The results of Deep ANN can be seen in Figure 6. The results showed that Deep ANN, in all cases, improved the accuracy of estimation. For canola dry biomass and LAI, the model delivered the RMSE of 25.8 g/m$^2$ and 0.525 m$^2$/m$^2$, respectively (Figure 6a,b). The results of deep ANN clearly showed improved canola dry biomass and LAI. For corn dry biomass and LAI, the deep ANN provided the RMSE of 54.43 g/m$^2$ and 0.273 m$^2$/m$^2$, respectively (Figure 6c,d). For both corn LAI and dry biomass, deep ANN improved the retrieval accuracy. Finally, for soybean dry biomass and LAI, the deep ANN provided the RMSE of 4.95 g/m$^2$ and 0.211 m$^2$/m$^2$, respectively (Figure 6e,f). Besides, the deep ANN slightly improved the estimation's accuracy for both LAI and dry biomass. A minimal amount of saturation can be seen in the high biomass for soybean' dry biomass (approximately higher than 60 g/m$^2$).

*4.5. Discussion*

For the last decades, remote sensing satellite SAR and optic data's progress provides an environment for further research on crop biophysical parameters. MLAs showed significant potential in broad areas; utilizing these methods recently has grown to solve remote sensing problems. Crop biophysical parameters are vital parameters for crop monitoring, crop stress assessments, crop growth model, to name but a few. Identify the number of train and test samples, the best value for each parameter in tuning MLAs' hyperparameters, and many other things that are not mentioned here, are the reasons that we need to compare several MLAs to determine the best approach to estimate target parameters. In this study, we focused on the potential of four MLAs to assess two crop biophysical parameters. XGB is a new method that is used in the fields related to crop parameter estimation. Information during crop growth duration is available from UAVSAR data. In general, for all three crops, a combination of UAVSAR polarimetric features and spectral VIs have a better performance to estimate crop biomass and LAI. the estimation accuracy of regression models UAVSAR L-band polarimetric features showed great potential in retrieving soybean dry biomass and LAI. For canola and corn LAI and dry biomass, generally, the accuracy of estimation using optical VIs was better than SAR polarimetric features in each regression model.

Considering other research works, Reisi-Gahrouei, et al. [44] achieved RMSE of 56.55 g/m$^2$ and R = 0.72 for canola's dry biomass using decomposition UAVSAR L-band data. Besides, they achieved an RMSE of 13.48 g/m$^2$ and R = 0.82 for soybean's dry biomass. In another study, Reisi Gahrouei, et al. [22] achieved 25.22 g/m$^2$ for canola, 88.13 g/m$^2$ for corn, 5.91 g/m$^2$ for soybean using spectral VIs extracted from RapidEye optical data. Their model delivered RMSE of 0.59 m$^2$/m$^2$ for canola, 0.27 m$^2$/m$^2$ for corn and 0.21 m$^2$/m$^2$ for soybean, a combination of UAVSAR L-band data and spectral VIs improved the soybean and canola dry biomass estimation in our study. Mandal, et al. [76] used various methods to estimate wet biomass and PAI of soybean and wheat. They achieved RMSE between 0.73 to 1.21 g/m$^2$ for wheat's wet biomass. As well, their results for wheat PAI were between 0.83 to 1.48 m$^2$/m$^2$. The best soybean wet biomass and PAI results were 0.34 g/m$^2$ and 0.72 m$^2$/m$^2$, respectively. We achieved the RMSE of 4.95 g/m$^2$ and 25.80 g/m$^2$ for soybean and canola dry biomass, respectively. Our model also provided the RMSE of 0.211 m$^2$/m$^2$ for soybean LAI, 0.273 m$^2$/m$^2$ for corn LAI, and 0.525 m$^2$/m$^2$ for canola LAI. Our model amazingly improved the accuracy of LAI estimation, especially for corn.

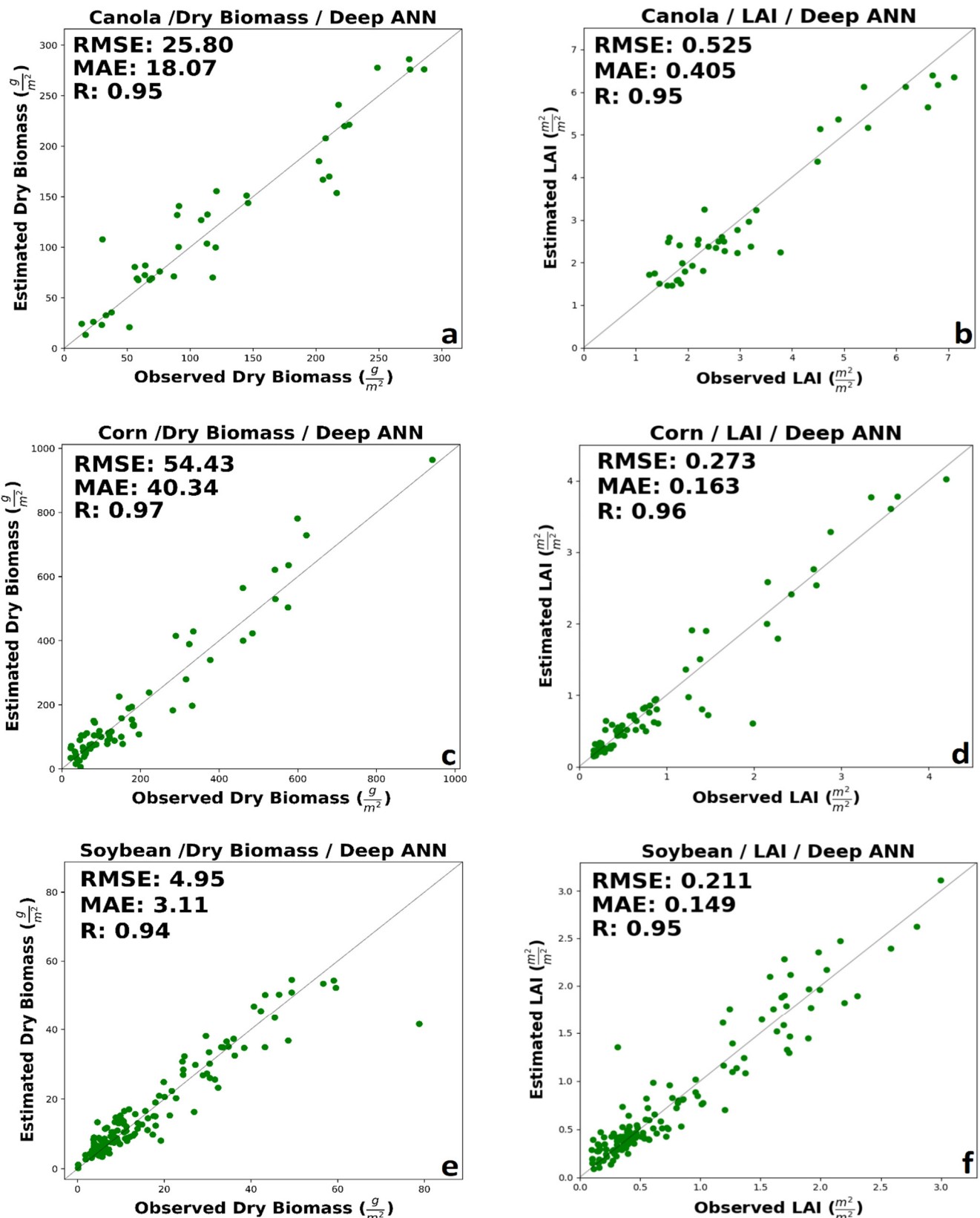

**Figure 6.** The results of Deep ANN for (**a**) canola's dry biomass, (**b**) canola's LAI, (**c**) corn's dry biomass, (**d**) corn's LAI, (**e**) soybean's dry biomass, and (**f**) soybean's LAI.

## 5. Conclusions

Biomass and LAI are two critical parameters in the crop growth model and crop monitoring. This paper assessed four MLAs' potential to estimate dry biomass and LAI of three crops, including soybean, corn, and canola. In situ measurements have been collected during the SMAPVEX-12 campaign over Manitoba, Canada. Several polarimetric features were extracted from UAVSAR data. Besides, various spectral VIs were extracted from RapidEye optical data. Correlation for all features was calculated; also, RF feature importance for each feature was obtained. Finally, the correlation with an absolute value of 0.9 was considered, and the feature with low importance and high correlation was removed. The remaining features were incorporated into machine learning regression models. The results showed that the integration of SAR polarimetric and spectral VIs better estimate dry biomass and LAI. Besides, XGB showed great potential in assessing crop biophysical parameters. For LAI, RMSE was reported as 0.557 $m^2/m^2$ for canola, 0.298 $m^2/m^2$ for corn, and 0.233 $m^2/m^2$ for soybean. Also, RMSE was reported for dry biomass as 29.45 $g/m^2$ for canola, 26.29 $g/m^2$ for corn, 5.00 $g/m^2$ for soybean. In addition, the results of deep neural networks were 0.525 $m^2/m^2$, 0.273 $m^2/m^2$, and 0.211 $m^2/m^2$ for canola, corn, and soybean LAI, respectively. The results of deep neural networks were 25.80 $g/m^2$, 57.97 $g/m^2$, 5.00 $g/m^2$ for canola, corn, and soybean dry biomass, respectively.

**Author Contributions:** H.B., S.H., S.M., and M.M.; methodology, H.B. and O.R.-G.; data processing, H.B. and M.M.; programming and implementation; H.B., S.H. and M.M.; validation, H.B., S.H., M.M., A.S. and S.M.; formal analysis: H.B., S.H., A.S., S.M. and M.M.; investigation, H.B. and S.H.; resources, S.H., M.M.; data curation, H.B., S.H. and M.M.; writing—original draft preparation, H.B., S.H., A.S., S.M. and M.M.; writing—review and editing, H.B.; visualization. All authors have read and agreed to the published version of the manuscript.

**Funding:** This research received no external funding.

**Institutional Review Board Statement:** Not applicable.

**Informed Consent Statement:** Not applicable.

**Data Availability Statement:** The SMAPVEX12 ground sampling data used in this study is available on https://smapvex12.espaceweb.usherbrooke.ca/ (access on 13 June 2021).

**Acknowledgments:** The authors thank the SMAPVEX12 team, Agriculture and Agri-Food Canada, for providing the in-situ measurements. Moreover, we truly appreciate the NASA Jet Propulsion Laboratory (JPL) for providing UAVSAR data.

**Conflicts of Interest:** The authors declare no conflict of interest.

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
