# Peer review of "Deep Learning-Based Estimation of Crop Biophysical Parameters Using Multi-Source and Multi-Temporal Remote Sensing Observations"

_agronomy, doi:10.3390/agronomy11071363_

Round 1

Reviewer 1 Report

This manuscript presents a study where satellite imagery is used in different machine learning algorithms to predict leaf area index (LAI) and biomass. Although timely and important from an agricultural perspective, this study fails to differentiate itself from previous studies that have also applied machine learning techniques to map various crop parameters. The authors provide a long list of past publications but fail to present their unique take on the subject. In addition to this shortcoming, there are also other matters that need to be addressed.

Major Issues:

  • The motivation behind this study and the approaches taken is unclear.
  • Much more time/space needs to be used to properly explain the datasets used, the processing that it underwent, and most importantly the feature engineering used for each of the ML approaches. Were inputs for each ML algorithm prepared in the same way? The methods section is confusing. ML can be a confusing subject and utilizing many ML approaches requires careful consideration to lay out the details in a concise and appropriate manner that is clear to the reader. Besides more detail within the text, a table defining features (inputs) and labels (outputs) for each model may be helpful (including percentages of training/testing).
  • The results section is simply a list of what can be found in the figures.
  • The discussion section follows the results section; it is just a list of findings/statistics. There is little within the manuscript that attempts to explain the results.

Throughout manuscript:

  • Transition between passive and active tense, sometimes in one sentence. Please be consistent throughout.

Line 47: Sentence should read: “including dry biomass”. Omit the “and”

Line 85: “models used physical laws”. Change to present tense or remain consistent with entire introduction.

Line 92: “corn, and canola using high-resolution”

Line 87-105: This paragraph is simply an exhaustive list of past research endeavors. I would suggest re-phrasing in such a way that adds to the manuscript. For example, how past research has led to the problem that will be addressed in the current study.

Line 106-120: I believe that there needs to be more of a distinction between past research and what is being proposed in the current project. What makes this approach/manuscript novel?

Line 149: Table 1 is confusing. What are the headers ‘Instrument’ and ‘UAVSAR’ referring to? For example, ‘Frequency’ and ‘Polarizations’ are not instruments. Confusing format for the reader.

Line 159: Why is RVI being introduced? Where will is be used? This paragraph needs more clarification/details.

Line: 174: MLAs not MALs

Line 247: I thought the dataset was split 2/3 for train, 1/3 for test? Is 20% just for ANN?

Figure 5: RMSE is not a good comparison across the crops types as each crop will have characteristic values of LAI. I suggest adding a normalized RMSE to this plot or not show Canola, Corn, and Soybean next to one another as the comparison is misleading.

Reviewer 2 Report

Few suggestions for the paper.

  • Parameter tunning for ML algorithm is missing. Please introduce it and refect its variance over the error.
  • EDA is missing, please include them in detail
  • Statistical testing is missing
  • Posthoc test is missing 
  • Please include the literature related to smart farming in the current state of the art, cite the papers if you find them relevant:
  • Das V., J.; Sharma, S.; Kaushik, A. Views of Irish Farmers on Smart Farming Technologies: An Observational Study. AgriEngineering 20191, 164-187. https://doi.org/10.3390/agriengineering1020013
  • Pivoto D, Waquil PD, Talamini E, Finocchio CP, Dalla Corte VF, de Vargas Mores G. Scientific development of smart farming technologies and their application in Brazil. Information processing in agriculture. 2018 Mar 1;5(1):21-32.

Round 2

Reviewer 1 Report

Thank you for addressing my suggestions/comments. I believe this is a much better manuscript. You have done a wonderful job of reformatting and I think this study shows the benefits of combining optical and radar earth observations. I simply ask the authors to read through the manuscript again to make sure clarity and sentence structure is complete throughout.

Author Response

We truly appreciate for accepting answers to the previous comments. We thoroughly and accurately investigated and checked the sentence structure and made sure about the sentences' clarity.